# Cry3Aa Toxin Is Not Suitable to Control Lepidopteran Pest *Spodoptera littoralis* (Boisd.)

**DOI:** 10.3390/plants11101312

**Published:** 2022-05-15

**Authors:** Oxana Skoková Habuštová, Zdeňka Svobodová, Dalibor Kodrík, František Sehnal

**Affiliations:** 1Institute of Entomology, Biology Centre, Czech Academy of Sciences, 370 05 České Budějovice, Czech Republic; svobodova@entu.cas.cz (Z.S.); kodrik@entu.cas.cz (D.K.); 2Faculty of Science, University of South Bohemia, 370 05 České Budějovice, Czech Republic

**Keywords:** *Spodoptera littoralis*, *Leptinotarsa decemlineata*, recombinant Cry3Aa, natural Cry3Aa, Superior NewLeaf, integrated pest management, biological control

## Abstract

The toxicity of the *Bacillus thuringiensis* (Bt) toxin Cry3Aa—originally used against the main potato pest, the Colorado potato beetle, *Leptinotarsa decemlineata*—was verified on this species and then evaluated against the Egyptian armyworm, *Spodoptera littoralis*, which is a pest of several economically important plants. Larvae of *S. littoralis* were fed a semi-artificial diet supplemented either with a recombinant or with a natural Bt toxin Cry3Aa and with the genetically engineered (GE) potato of variety Superior NewLeaf (SNL) expressing Cry3Aa. Cry3Aa concentration in the diet and the content in the leaves were verified via ELISA (enzyme-linked immunosorbent assay) and HPLC (high-performance liquid chromatography) during and at the end of the experiments. The biological effectiveness of the coleopteran-specific Cry3Aa with previous reports of activity against *S. littoralis* was tested on five different populations of *S. littoralis* larvae by monitoring 13 parameters involving development from penultimate instar, weight, the efficiency of food conversion to biomass, ability to reproduce, and mortality. Although some occasional differences occurred between the Cry3Aa treatments and control, any key deleterious effects on *S. littoralis* in this study were not confirmed. We concluded that the Cry3Aa toxin appears to be non-toxic to *S. littoralis*, and its practical application against this pest is unsuitable.

## 1. Introduction

One of the environmentally friendly methods used to reduce insect pest populations is the practical utilisation of the insecticidal crystal protein (Cry) that occurs naturally in the soil bacterium *Bacillus thuringiensis* (Ber.) (Bt). Cry toxins are usually applied via spraying or in genetically engineered (GE) plants. Cry toxins kill host cells and thus allow Bt germination in dead arthropods. Cry toxins are intestinal pore-forming δ-endotoxins that, after activation by host proteases in the midgut, interact with receptors on the midgut epithelium. For example, in Lepidoptera, aminopeptidase N (APN) receptors, cadherin-like receptors, and ATP binding cassette (ABC) protein family function as toxin receptors for Cry1A. They are involved in the cleavage of the amino-terminal end, including the helix, and the formation of a pre-pore oligomer of Cry toxin, which leads to membrane insertion and pore formation. The pore formation results in osmotic cell lysis or else activation of the oncotic cell death pathway [1,2]. Because of their interaction with greatly diversified receptors, Cry toxins are highly specific to certain species of the insect orders Lepidoptera, Coleoptera, Hymenoptera, Diptera, Orthoptera, and Mallophaga, and also to Nematoda, Acari, and Protozoa [3]. However, some Cry toxins have an expanded spectrum of action to two or more taxonomic categories. For example, Cry1B is one of those that present a remarkable activity against larvae of Lepidoptera, Diptera, and Coleoptera [4].

Cry1 act on lepidopteran pests, and therefore Cry1Ab suppresses *Spodoptera littoralis* (Boisd.) (Lepidoptera: Noctuidae) [5,6,7]. On the other hand, Cry3 toxins are specific to coleopteran species [8,9,10]; Cry3Aa is used against potato pest *Leptinotarsa decemlineata* (Say) (Coleoptera: Chrysomelidae). Interestingly, in some previous experiments, a certain cross-activity of the Cry toxins among the insect orders was recorded; for example, the Cry3Aa was found to affect non-target lepidopterans, namely, the early instars of *Acherontia atropos* (L.), *Manduca sexta* (L.) (both Lepidoptera: Sphingidae), and *Autographa gamma* (L.) (Lepidoptera: Noctuidae) [11]. Later, it was reported that the Cry3Aa toxin also reduced larval growth of *S. littoralis* when fed a Cry3Aa-expressing potato, and larval growth, pupal size, and adult fecundity when fed Cry3Aa in a semi-artificial diet [12,13,14]. Therefore, we decided to review and extend the data on the effect of the Cry3Aa toxin on five different populations of *S. littoralis*, two of which should have high sensitivity to insecticides.

It is well known that *S. littoralis* is a polyphagous and economically important pest of many cultivated plants in the Mediterranean region [15]. It is an A2 quarantine pest in the European Union (EU) with occasional occurrence in Central Europe, where its permanent existence is not yet possible. However, climate change could alter its distribution, and Central European potatoes and other crops may be in danger by this novel pest [16]. New agents against this pest applicable in integrated pest management (IPM) and organic farming would be appreciated by farmers because pressure to utilise sustainable agriculture practices is considerable worldwide.

We used natural and recombinant Cry3Aa toxins applied in a semi-artificial diet and the GE potato Superior NewLeaf™ (MONSANTO Technology LLC, St Louis, MO, USA), which expresses Cry3Aa. This potato is resistant to *L. decemlineata* with a simultaneous absence of effects on beneficial arthropods such as lady beetles and carabid beetles in laboratory and field studies [17,18,19].

The main objective of the present study was to investigate and extend the existing data on the efficacy of various forms of Cry3Aa (recombinant, natural, and expressed in GE potato) on the pest *S. littoralis* with possible implications in IPM.

## 2. Results

### 2.1. Cry3Aa Content

The relative amount of Cry3Aa in the working solutions was determined by semiquantitative RP HPLC (Figure 1). The results suggested a higher (about 7.5-fold) level of the recombinant protein than the natural protein in the corresponding solutions. The results obtained using the RP HPLC were supported by ELISA. These showed that the working solutions of recombinant and natural Cry3Aa contained 3.418 µg/mL and 279 µg/mL protein, respectively. These amounts were stable and constant until the end of the experiments. The diet used in the experiments contained an amount of Cry3Aa that was based on the concentrations determined by ELISA. Further, the content of the Cry3Aa in potato leaves used for bioassays 1 and 3 ranged from 1.31 to 1.96 µg/g Cry3Aa of fresh weight.

### 2.2. Effect of Cry3Aa on Survival of L. decemlineata

The results showed that the effect of natural Cry3Aa was more pronounced than effect of recombinant Cry3Aa (Figure 2A,B). Thus, lower concentrations of natural Cry3Aa were utilised to determine the LC_50_ and LC_90_. LC_50_ was determined to be 1.8 µg/g (95% confidence limits: 0.78–2.74) and 0.1 µg/g (95% confidence limits: 0.03–0.23) for recombinant and natural Cry3Aa, respectively. LC_90_ was calculated to be 8.1 µg/g (95% confidence limits: 4.91–33.68) and 1.2 µg/g (95% confidence limits: 0.56–4.41) for recombinant and natural Cry3Aa, respectively. The values of LC_50_ and LC_90_ for the natural Cry3Aa was about 18 and 6.8 times lower than their recombinant forms, respectively. The effectiveness of Cry3Aa toxins was different for various concentrations (Log-rank test: recombinant Cry3Aa: χ2 = 105.3, df = 6, *p* < 0.0001; natural Cry3Aa: χ2 = 282.9, df = 9, *p* ≤ 0.0001, results of post hoc tests in Appendix A). The effect of Cry3Aa expressed in leaves of GE potato SNL was also evident from the second day. Survival curves were significantly different between GE potato SNL and control (Log-rank test: χ2 = 93.9, df = 1, *p* ≤ 0.0001). On the fifth day, more than 90% of *L. decemlineata* on leaves of GE potato SNL were dead (Figure 2C). Compared with that, we estimated from Figure 2A,B that recombinant and natural form in same concentrations caused approximately 20% and 38% mortality, respectively. On the basis of the results of bioassay 1 (LC_90_), in bioassay 2, we worked with a concentration of 8 µg/g of Cry3Aa in a semi-artificial diet.

### 2.3. Effect of Cry3Aa on Larval and Pupal Mortality of S. littoralis

In bioassays 2 and 3, certain differences in larval and pupal mortalities between Cry3Aa-treated *S. littoralis* and corresponding controls were recorded (Table 1 and Table 2). However, the difference was mostly insignificant. In the NRC population within bioassay 2 (Table 1 and Appendix A), the overall test (*p* = 0.046 for larvae, and *p* = 0.036 for pupae) indicated statistical significance; however, post hoc tests did not reveal any specific difference between treatments. Further, in bioassay 3 within the SF population, pupal mortality was significantly higher (about 2.4-fold) in the control compared to GE potato SNL feeding (*p* = 0.011) (Table 2 and Appendix A).

### 2.4. Sublethal Effects of Cry3Aa on S. littoralis

Within bioassay 2 in the NRC population, recombinant Cry3Aa treatment caused an increase in body weight and a difference in maximal body weight (1.1 times for both) in comparison with those in the natural Cry3Aa and control treatments (*p* ˂ 0.001), but the body weight gain was similar between treatments (Table 1 and Appendix A, Figure 3A). In the SE population, recombinant Cry3Aa treatment produced a higher weight increment (*p* = 0.029) and maximal body weight (*p* = 0.025) than natural Cry3Aa. Both parameters were 1.1 times higher than those in the natural Cry3Aa treatment, which was also 1.2 times lower than in the control (*p* = 0.002) for both parameters. Similarly, the body weight gain was highest in the control, followed by the recombinant and natural Cry3Aa treatments (Figure 3B). The length of the fifth instar in the SE population fed recombinant Cry3Aa was 1.2 times shorter than those fed natural Cry3Aa, and the control (*p* = 0.003). In the NRC population, the pupal weight of the recombinant Cry3Aa treatment was 1.1 times higher than in the other treatments (*p* = 0.017 compared with the control, and *p* = 0.047 compared with the natural Cry3Aa treatment), and the length of the pupal stage was 1.2 times longer than the other two treatments (both tests: *p* < 0.001). Moreover, the length of the pupal stage in the natural Cry3Aa treatment was 1.2 times longer than natural Cry3Aa (*p* = 0.045). In the SE population, the pupal weight of the control was 1.1 times higher than in both the natural and recombinant Cry3Aa treatments (*p* = 0.035). In the SF population, the length of the pupal stage was 1.1 times longer in the natural than in the recombinant Cry3Aa treatments (*p* = 0.033). Other results of the statistical comparison were not significantly different (Table 1 and Appendix A, Figure 3C). The highest hatching rate was found in the SE population, followed by the SF and NRC populations.

Although the curves for body weight gain in bioassay 3 looked similar for both treatments, in the NRC population, body weight increased more intensively at the end of sixth instar in the GE potato SNL treatment than in the control (Figure 4B). The opposite trend at the end of the sixth instar was recorded in the SF population (Figure 4E). In the SE population, the body weight of the GE potato SNL treatment increased slower than in the control, but maximal body weight was higher in the GE potato SNL treatment than in the control (Figure 4D). In the NRC populations, the sixth instar was 1.1 times longer in the control than those in the GE potato SNL (*p* = 0.031). Similarly, in the SF population, the fifth instar was 1.3 times longer in the control than those in the GE potato SNL (*p* = 0.029), and the length of the prepupal stage of the GE potato SNL treatment was 1.1 times longer than in the control (*p* ˂ 0.001). On the contrary, the pupal stage was 1.1 times longer in the control (*p* = 0.018). In the NRC and SE populations, the pupal stage was longer in the GE potato SNL treatment than in the control by 1.1 times (*p* ˂ 0.001, both). Other results of the statistical comparison were not significantly different (Table 2 and Appendix A, Figure 4). The highest hatching rate was identified in the CU population, followed by the NRC, SE, N, and SF populations.

## 3. Discussion

In this study, we sought to characterise the effect of a Cry3Aa toxin on the lepidopteran pest *S. littoralis*. Three types of the Cry3Aa toxin were tested—recombinant, natural, and that expressed by GE potato SNL.

### 3.1. Bioassay 1—Effect of Cry3Aa on L. decemlineata

We verified the efficacy of three forms of Cry3Aa on first instar *L. decemlineata* larvae. The results confirmed the high effectiveness of tested toxins. Interestingly, the recombinant and natural Cry3Aa toxins differed in their efficacy. Cry3Aa expressed in GE potato SNL showed the highest efficiency. The distinction could be caused by any difference in protein three-dimensional arrangement given by organisms in which they were synthetised. Moreover, the recombinant and natural Cry3Aa in solution could be less efficient because they could be subject to varying degrees of degradation in comparison with Cry3Aa in GE potato SNL leaves that was permanently synthetised in leaves [11,20]. However, variation in LC_50s_ of Cry3Aa has already been subject of research and extensive discussion [21,22]. Generally, our mean LC_50_ values are approximately 10 times lower than mean value obtained by Robertson et al. [22], but they are in the range of values they determined. The results of bioassay 1 showed that all examined forms of Cry3Aa toxin were active and effective against *L. decemlineata*. These results are not surprising, because the efficacy of the Cry3Aa toxin against *L. decemlineata* is generally known, e.g., [17,23,24,25], and Cry3Aa toxin is used in agricultural practice in the form of spray (organic farming) or incorporated in GE potatoes [26].

### 3.2. Bioassay 2—Effect of Cry3Aa in Semi-Artificial Diet on S. littoralis

First, we analysed the effect of Cry3Aa applied in a semi-artificial diet. We selected Cry3Aa toxin concentration on the basis of its efficacy on *L. decemlineata* in bioassay 1. We assumed lower activity on *S. littoralis*. We tested penultimate larval instar of two insecticide-sensitive populations, SE and SF, and one common population, NRC. Significant differences in several parameters were found. These results suggested certain Cry3Aa activities. Haider and Ellar [27] suggested that the specific endotoxin-binding receptors in the gut were not necessarily a precondition of a toxic effect. Thus, Lepidoptera eventually would not need a suitable receptor, but suitable protease would be necessary to activate Cry3Aa [11]. Furthermore, Hussein et al. [12] claimed that Cry3Aa ingested by the *S. littoralis* larvae could cause reparable injuries in their midgut epithelium, as was described for *Manduca sexta* after ingestion of suspended crystal endotoxin from *B. thuringiensis* ssp. kurstaki HD1 [28]. This phenomenon could influence the food intake and reduced food consumption, which causes smaller biomass increments and longer development, but manifestation of sublethal effects is limited after switching to non-toxic food [28]. This phenomenon may explain the longer pupal stage for the NRC and SF population in recombinant and natural Cry3Aa treatment, respectively. Nevertheless, the fifth instar of recombinant Cry3Aa treatment in the SE population was shorter than in other treatments. We recorded higher values of body weight parameters in the NRC population for recombinant Cry3Aa in comparison with other treatments, but conversely lower values of body weight parameters in SE population for natural Cry3Aa in comparison with other two treatments. These results suggest that this phenomenon is not fully applicable to our results. Furthermore, it seems that it is possible that pupal weight is not affected even in lepidopteran species sensitive to consumed Bt toxin, although mortality and prolonged development was recorded [29]. Nevertheless, reports about heavier pupae are also available [30].

We used two different Cry3Aa toxins. Although, the different methods of their preparation might play a crucial role in Bt toxin efficiency [11], in contrast to the effect on *L. decemlineata*, we did not determine any constant difference in observed parameters that would imply dissimilarity in Cry3Aa toxicity. However, there is another phenomenon that can substantially affect the efficacy of Cry toxin—the age of the tested insect [11]. The high susceptibility of the first larval instars of *S. littoralis* to lepidopteran-specific Cry toxins and decline in its effectiveness in following larval development have been demonstrated several times [11,31]. Whilst this has been satisfactorily explained [32], there are also reports of adverse effects of lepidopteran-specific Cry toxins at all stages of *S. littoralis* development [33,34,35,36].

Results from bioassay 2 showed that the concentration of 8 µg of Bt Cry3Aa toxin per gram of the semi-artificial diet, which is approximately five times higher than in GE potato SNL leaves, did not cause any evident and uniform effect on observed parameters of *S. littoralis.*

### 3.3. Bioassay 3—Effect of Cry3Aa Expressed in GE Potato SNL on S. littoralis

In bioassay 3, we tested the effect of Cry3Aa toxin expressed in GE potato SNL on selected populations of *S. littoralis* (sensitive SE and SF populations, and N, NRC, and CU long-term laboratory populations).

Primarily, the results showed occasional significant differences in length of larval, prepupal, and pupal stages and mortality and body weight between SNL potato and the corresponding control. Nevertheless, the actual differences were not dramatic. Other parameters including female fecundity and hatchability of progeny were not affected by the Cry3Aa treatment. In addition, as in bioassay 2, all results showed no clear tendency to indicate positive or negative effects of the Cry3Aa, because the differences were recorded in both directions (higher/lower, longer/shorter) for GE potato SNL and control. Anyhow, in this assay, we tested several populations of *S. littoralis* and received a sometimes significant but generally low effect of the Cry3Aa toxin. Thus, we assume we can generalise our results to other *S. littoralis* populations and conclude that GE potato SNL is not significantly resistant to *S. littoralis*.

### 3.4. Larval and Pupal Mortality (Survival) of S. littoralis in Cry3Aa Treatments

Mortality is a basic aspect for assessing the deleterious effect of any toxin. Differences in mortality levels after the Cry3Aa treatments within the populations were not significant. Thus, it is evident that mortality was not dependent on the applied insecticidal Cry3Aa toxin but was likely affected by any other parameter(s). It is interesting to note that mortality in bioassay 3, where the GE potato SNL was used, was in the most cases higher than in bioassay 2, where only a semi-artificial diet was used. We can speculate that switch of diet may play a role (basic cultures were kept on a semi-artificial diet); this is supported by relatively high mortalities in controls (see Table 2). Nevertheless, it is peculiar that the sensitive SF population showed higher mortality (for pupae, even significantly higher) in controls than that in treated insects within bioassay 3; we are at present unable to offer any satisfactory explanation for this, but some connection with the diet switch cannot be excluded. In contrast, *S. littoralis* is a polyphagous species and should tolerate wider spectrum of diets; therefore, we decided to start our experiments immediately after the populations were delivered to our laboratory. Additionally, we wanted to preserve their features and not affect them by breeding in our conditions. Furthermore, it is usual that tested species are exposed to unusual food sources in the assessment of Cry toxins and GE crops expressing Cry toxins without becoming accustomed to new food, e.g., [19,37], but it is important to separate the effect of Cry toxin and nutritional stress [38].

### 3.5. Sublethal Effects of Cry3Aa on Different S. littoralis Populations

No differences were found in four of the tested parameters after the Cry3Aa treatments. Nevertheless, there were significant differences in eight recorded parameters, namely, in weight increment, body weight gain, maximal body weight, pupal weight, length of the fifth and sixth instar, and length of prepupal and pupal stage between both Cry3Aa treatments and the controls in both bioassays. However, as mentioned above for mortality, the actual differences of studied parameters were just slight—both positive and negative—and thus it is impossible to specify any constant effect of Cry3Aa from these results. In contrast, we can speculate that some of these differences might be explained as a consequence of Cry3Aa ingestion, which could cause reparable effect of midgut epithelium and slowdown in development [29,30]. We cannot exclude the effect of the food switch or natural variability of tested individuals.

Natural variation is a numerical difference in response that is detected each time a bioassay is repeated with one genetic group (population in our case) either within a single generation or population [21]. As a result of natural variation, responses of a tested group at any one time will therefore never be the same as responses of another group tested either at the same or different time [22]. Robertson et al. [21] and recently also Chen et al. [38] demonstrated that once variation for cohorts or generations are assessed, realistic conclusions about values outside the range of natural variation can be drawn. For this reason, in any study of population sensitivity, responses of any species must be estimated with unselected cohorts within a population or for several generations [21] as we did.

## 4. Conclusions

Our study did not show a deleterious effect of Cry3Aa on the pest *S. littoralis*. We explained the observed differences in the parameters between the Cry3Aa and control treatments primarily as a result of food switch and natural variation. Thus, according to our results, Cry3Aa toxin is not suitable for the control of *S. littoralis* populations in any form, and therefore we do not recommend using it as a natural insecticide against *S. littoralis* in IPM and organic farming.

## 5. Materials and Methods

### 5.1. Culture of Leptinotarsa Decemlineata

The adults and larvae of *L. decemlineata* were collected from the potato plants in the vicinity of Biology Centre, Czech Academy of Sciences, České Budějovice, Czech Republic (48.97417 N, 14.44867 E), in several consecutive series. The collected *L. decemlineata* was placed inside the fine mesh cage (100 × 50 × 50 cm). Culture was maintained in controlled greenhouse conditions (25 °C, 75% relative humidity, photoperiod of 16 h of light/8 h of dark). Culture was supplemented daily with fresh potato plants of the variety Magda. Potato plants of the variety Magda were obtained in the form of tubers and tissue cultures (tiny plants) from the Potato Research Institute, Havlíčkův Brod, Czech Republic. Plants were grown in a pot with a diameter of 21 cm and a volume of 4 L, watered regularly, and kept in the same conditions as a *L. decemlineata* culture.

### 5.2. Cultures of Spodoptera Littoralis

Five populations of *S. littoralis* were obtained from different localities and were kept in different conditions before the experiments started.

**Population N**: Larvae were collected in the vicinity of Cairo, Egypt, and kept in the National Research Centre, Giza, Egypt, for many years. This population was obtained by our laboratory several years ago and kept on a Manduca–Heliothis Premix diet (Stonefly Industries Inc., Bryan, TX, USA).

**Population NRC**: The population was obtained from the National Research Centre, Giza, Egypt, where it was kept for many years on the castor *Ricinus communis* (Euphorbiaceae) leaves with occasional feeding of some generations on the agar-bean semi-artificial diet (see below). To maintain the culture in our laboratory, the Manduca–Heliothis Premix diet was used.

**Population CU**: This population was obtained from the Faculty of Agriculture, Cairo University, Egypt, where it was kept on the *R. communis* leaves. Culture was kept on the Manduca–Heliothis Premix diet in our laboratory.

**Population SE**: This population sensitive to insecticides was received from the Central Agricultural Pesticides Laboratory, Agricultural Research Centre, Giza, Egypt, where it was maintained on the agar-bean semi-artificial diet (see below) for many years. To increase the vigour of the progeny, one generation per year was fed *R. communis* leaves. In our laboratory, the same agar-bean semi-artificial diet was used.

**Population SF**: The sensitive population was obtained from the French National Institute for Agricultural Research (INRA), Versailles, France, where they were reared on a diet based on soya powder and maize bran (pinole) with antibiotics (see below). In our laboratory, larvae were kept on the same diet.

In our laboratory, all *S. littoralis* cultures were kept at 25 °C at a photoperiod of 16:8 h, and they were fed ad libitum. Experiments were carried out with the first generation of larvae that were delivered to our laboratory.

### 5.3. Semi-Artificial Diets

The recipe for semi-artificial agar diet for *L. decemlineata* is available in S1: D. The recipe for semi-artificial agar bean diet for *S. littoralis* is given in S1: E. The recipe for soy powder and corn bran diet for *S. littoralis* is described in S1: F. The Manduca–Heliothis Premix diet was prepared from commercially available powder (Stonefly Industries Inc., TX, USA) according to the instructions in the manual, but potassium bicarbonate buffer pH 10.6 was used instead of water. The use of buffer shifted diet pH from 5.1 (prepared with distilled water) to 8.2, which is more favourable for Cry3Aa stability. Diets were stored in the refrigerator (4 °C) for up to one month.

### 5.4. Origin of Cry3Aa Toxins

The recombinant Cry3Aa crystals produced in *Escherichia coli* was provided by MONSANTO Technology LLC. The crystals were dissolved in 0.1 M potassium bicarbonate buffer (pH 10.6) to prepare working solution, centrifuged, stored in a refrigerator, and used within two weeks.

The purified natural Cry3Aa crystals from *B. thuringiensis* ssp. tenebrionis were provided by Igor A. Zalunin (Scientific Research Institute for Genetics and Selection of Industrial Microorganisms, Moscow, Russia, [39]). Working solutions were prepared with 0.05 M potassium bicarbonate buffer (pH 10.6, 0.001 M EDTA), centrifuged, stored in a refrigerator, and used within two weeks.

GE potato variety Superior NewLeaf™ (SNL) expressing Cry3Aa was obtained from MONSANTO Technology LLC, St. Louis, MO, USA, The GE potato SNL plants and their near-isogenic unmodified variety Superior were grown according to standard techniques [14].

### 5.5. Quantification of Cry3Aa Toxins

The relative amount of Cry3Aa in working solutions was verified by reversed-phase high-performance liquid chromatography (RP HPLC). Both Cry3Aa toxins (recombinant and natural) were dissolved in 0.11% TFA (trifluoroacetic acid) and analysed on the RP HPLC system by Clarity software (Data Apex version 8.0) with a Waters 2487 UV detector (wavelength 215 nm), using a Chromolith Performance RP-18e column 150−4.6 mm (Merck), solutions A and B (A—0.11% TFA in water; B—0.1% TFA in 60% acetonitrile), and a flow rate of 1 mL/min. The relative titre of the toxins was estimated from the areas of the corresponding HPLC peaks.

In another series of experiments, the levels of recombinant and natural Cry3Aa in working solutions, in a semi-artificial diet, and potato plants were checked by using the commercial enzyme-linked immunosorbent assay (ELISA) PathoScreen Complete Kit PSA 05900/0288 Bt-Cry3A (Agdia-Biofords, Evry Cedex, France) at the time of diet preparation and the end of storage of the working solutions. The assay was performed according to the manufacturer’s protocol. A positive control, supplied with an ELISA kit, was used to construct a standard curve with a twofold dilution series ranging from 0.16 to 20 ng/mL for potato leaves and stock solution, and 1.25 to 160 ng/mL for semi-artificial diet. The sensitivity threshold of the assay was 0.16 ng Cry3Aa per 1 g of fresh plant tissue and per 1 mL of both stock solution and 1.25 ng Cry3Aa per 0.1 g of semi-artificial diet. Absorbance was determined using an ELISA reader (Spectra MAX 340 PC, Molecular Devices, LLC., Sunnyvale, CA, USA) at 630 nm.

### 5.6. Bioassays

**Bioassay 1**—*L. decemlineata*: We verified the efficacy of tested Cry3Aa toxins on the larvae of susceptible coleopteran *L. decemlineata*. Freshly laid eggs were used. The eggs were transferred one by one from the potato leaf by a needle and entomological forceps and dipped individually for 1 s in 0.1% formaldehyde. Excess formaldehyde was removed by touching a filter paper, and eggs were transferred onto a sterile wet filter paper in a sterile glass/plastic Petri dish and incubated at 25 °C and a photoperiod of 16:8 h until larvae hatching. Mobile larvae not older than 30 h were put into a 48-well titre plate on a semi-artificial diet (S1: D) with different concentrations of Cry 3Aa toxin to investigate 50 and 90% lethal concentration (LC_50_ and LC_90_). In the case of testing effect of Cry3Aa expressed in GE potato SNL leaves, freshly hatched larvae were placed into a 48-well titre plate on the cut-out disk of control and GE potato SNL leaves. Potato disks were underlaid with moistened filter paper. The plates were tightly closed with a food foil (Saran wrap), punctured 3 times over each well with an insect pin (size 00), covered with a provided plastic lid, and kept at 25 °C and a photoperiod of 16:8 h. Mortality was recorded daily. The bioassay 1 was terminated in 8 days. For the exact number of larvae per treatment, see S2: G.

**Bioassay 2**—*S. littoralis* on a semi-artificial diet: Freshly moulted fifth (penultimate) instar larvae were selected from the NRC, SE, and SF populations. Larvae were divided into three treatments: a semi-artificial diet with natural Cry3Aa, a semi-artificial diet with recombinant Cry3Aa, and a control diet. Both recombinant and natural Cry3Aa toxins were administered in the Manduca–Heliothis Premix diet at a final concentration of 8 µg/g Cry3Aa in the diet. Larvae were kept separately, each in a Petri dish (9 cm in diameter); for exact number of larvae per treatment, see S2: H. Each experiment was repeated three times. Pupae were sexed and kept separately in plastic cups (4.5 cm diameter, 0.18 l volume) filled with two layers consisting of a 2 cm layer of fine sawdust and a 5 cm soil layer. Cups were sealed by netting until adult eclosion. The adults (1 ± 0.5 days old) were randomly paired, one of each sex, transferred into paper cylinders (10 cm high, 9 cm diameter), sealed on both sides with a Petri dish lid, and provided with the 10% honey solution (without added toxin). The experiment was terminated 10 days after the start of egg laying. The following parameters were monitored daily in each population: initial larval weight; body weight gain; ECI (efficiency of conversion of ingested materials; weight gain/(ingested diet—vapor) ∗ 100); maximum body weight; pupal weight; fifth instar length; sixth instar length; larval mortality; prepupal and pupal stage length; pupal mortality; number (no.) of laid eggs per female per day; no. of hatched eggs per female per day; and hatching rate (no. of laid/hatched eggs) per female per day. Larvae in bioassay 2 were maintained at the same temperature and light conditions as the stock cultures.

**Bioassay 3**—*S. littoralis*: Freshly moulted *S. littoralis* larvae of the fifth (penultimate) instar of populations N, NRC, CU, SE, and SF were divided into two treatments: feeding GE potato SNL and control with isogenic Superior that does not produce toxin. Larvae were placed individually in plastic cups (9 cm top diameter, 0.5 l volume), covered with netting, and fed daily fresh leaves placed in a small tube containing water; squares of cotton pads and aluminium foil prevented water leakage. For the exact number of larvae per treatment, see S2: I. The remaining procedures and monitoring parameters were the same as in bioassay 2.

### 5.7. Data Analysis

**Bioassay 1**: Log-rank (Mantel–Cox) tests with Bonferroni correction of the significance level of post hoc tests [40] were calculated to analyse the difference between survival curves of *L. decemlineata* on a diet with different concentrations of Cry3Aa. Probit analysis was applied for LC_50_ and LC_90_ calculations.

**Bioassay 2** and **3**: Analysis of covariance (ANCOVA) was used to eliminate the effect of sex included as a covariate in the analysis. ANCOVA was used to evaluate the data of initial larval weight, weight gain, and maximum body weight; fifth and sixth instar length; prepupal and pupal stage length; and pupal weight. In bioassay 2, where three treatments were compared (control, recombinant Cry3Aa, natural Cry3Aa), Tukey’s post hoc test followed significant tests to specify the results (between which treatments the difference was found). One-way ANOVA was used for ECI, the number of laid and hatched eggs per female per day. The chi-squared test was used for larval and pupal mortality. The chi-squared test for trend was used for the body weight gain during development. In bioassay 2, Bonferroni correction of significance level was applied in the chi-squared test and the chi-squared test for trend.

Data were analysed using PoloPlus (probit analysis, LeOra Software, Robertson et al. [41]), STATISTICA 8 for Windows (ANOVA, ANCOVA, StatSoft Inc., Tulsa, OK, USA) [42], and GraphPad Prism 5 (Log-rank test, chi-squared test, chi-squared test for trend, GraphPad Software Inc.) [43]. If not stated otherwise, a two-sided α-value of 5% was used to determine the level of significance. F-values were accompanied by degrees of freedom and degrees of freedom of the error (within-population degrees of freedom). On the basis of the Cochran C, Hartley, and Bartlett statistic, homogeneity of variances was confirmed, and normal approximation was applied. Chi-squared values were accompanied by degrees of freedom. Graphs were constructed in GraphPad Prism 5. Mean values were presented with standard deviation (mean ± SD).

## Figures and Tables

**Figure 1 plants-11-01312-f001:**
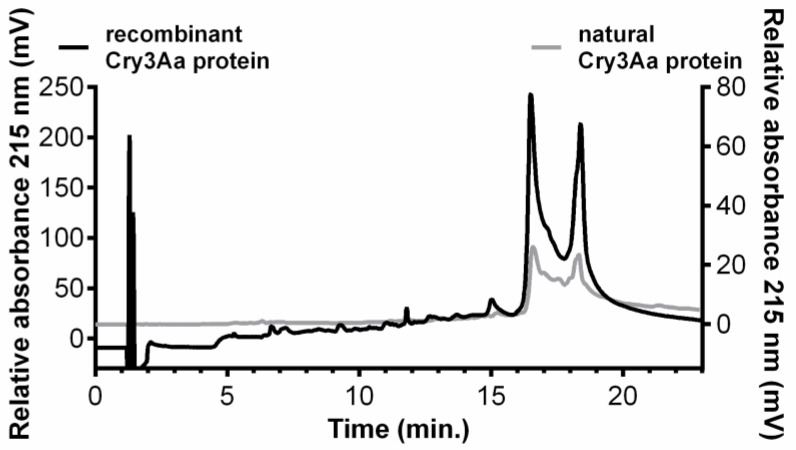
The RP HPLC elution profiles of recombinant Cry3Aa (left *y*-axis) and natural Cry3Aa (right *y*-axis) solutions (200 μL).

**Figure 2 plants-11-01312-f002:**
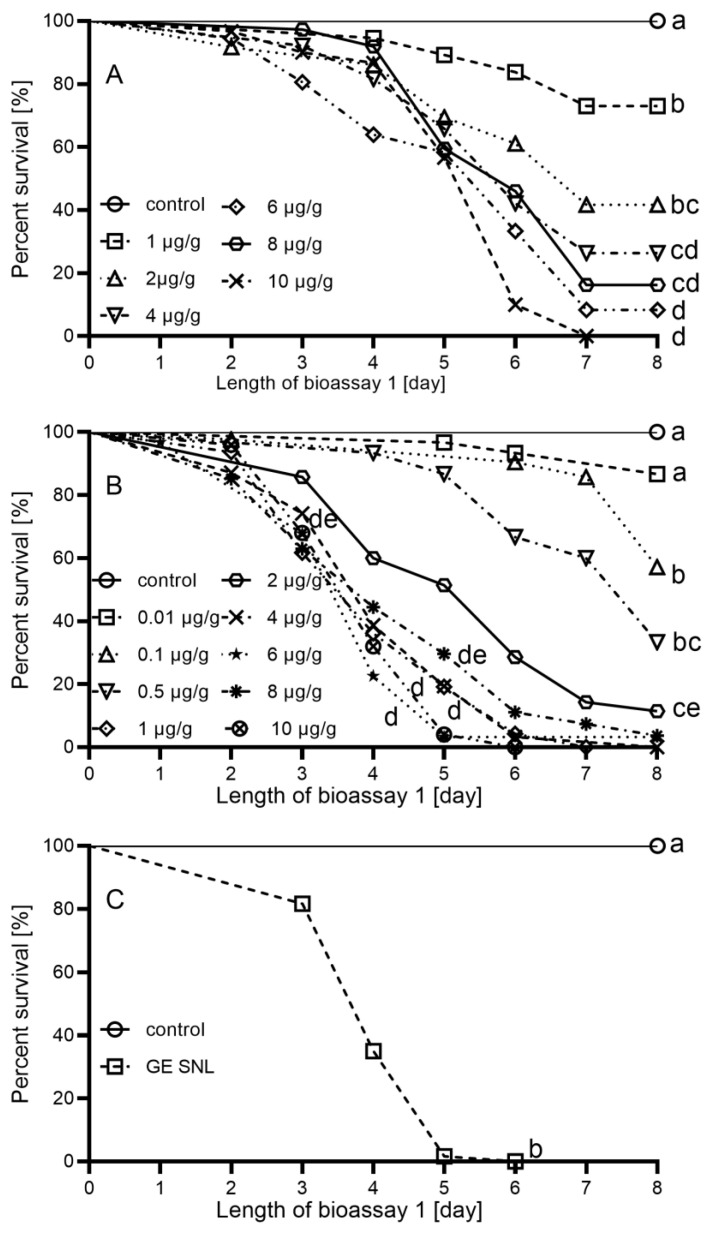
Survival of *L. decemlineata* larvae on the semi-artificial diet with different concentrations of recombinant (**A**) and natural (**B**) Cry3Aa, and on the leaves of GE potato SNL plants expressing Cry3Aa (**C**) in bioassay 1. The same letters denote non-significant differences, while different letters denote statistically significant differences in trend of survival between treatments. The values of statistical tests are available in Appendix A.

**Figure 3 plants-11-01312-f003:**
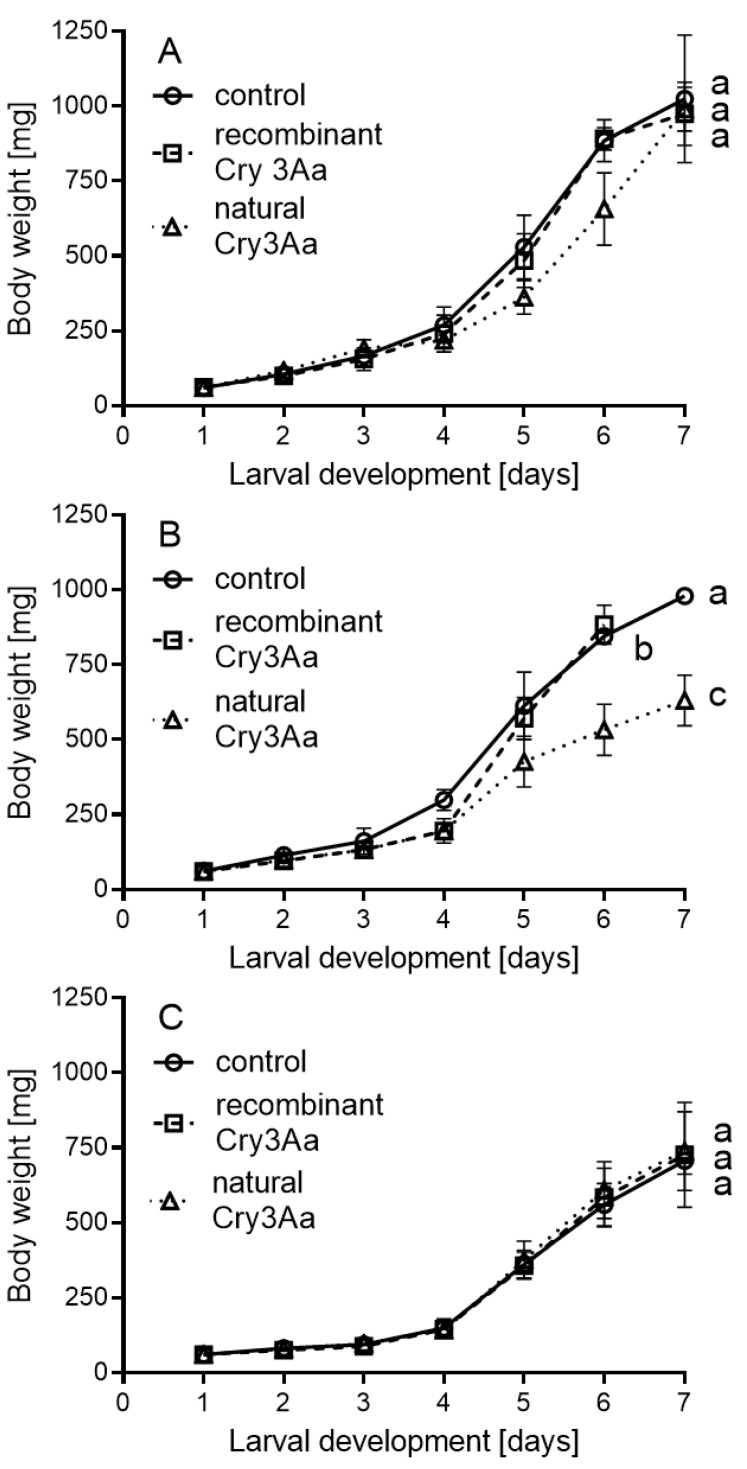
The body weight gain of *S. littoralis* (mean ± SD) larvae in NRC (**A**), SE (**B**), and SF (**C**) populations in bioassay 2 with 8 µg/g recombinant and natural Cry3Aa in a semi-artificial diet. The same letters denote non-significant differences, while different letters denote statistically significant differences. The values of statistical tests are available in Appendix A. N—population of *S. littoralis* reared in our laboratory; NRC—population of *S. littoralis* obtained from National Research Centre, Egypt; CU—population of *S. littoralis* obtained from Cairo University, Egypt; SE—sensitive population of *S. littoralis* obtained from Egypt; SF—sensitive population of *S. littoralis* obtained from France.

**Figure 4 plants-11-01312-f004:**
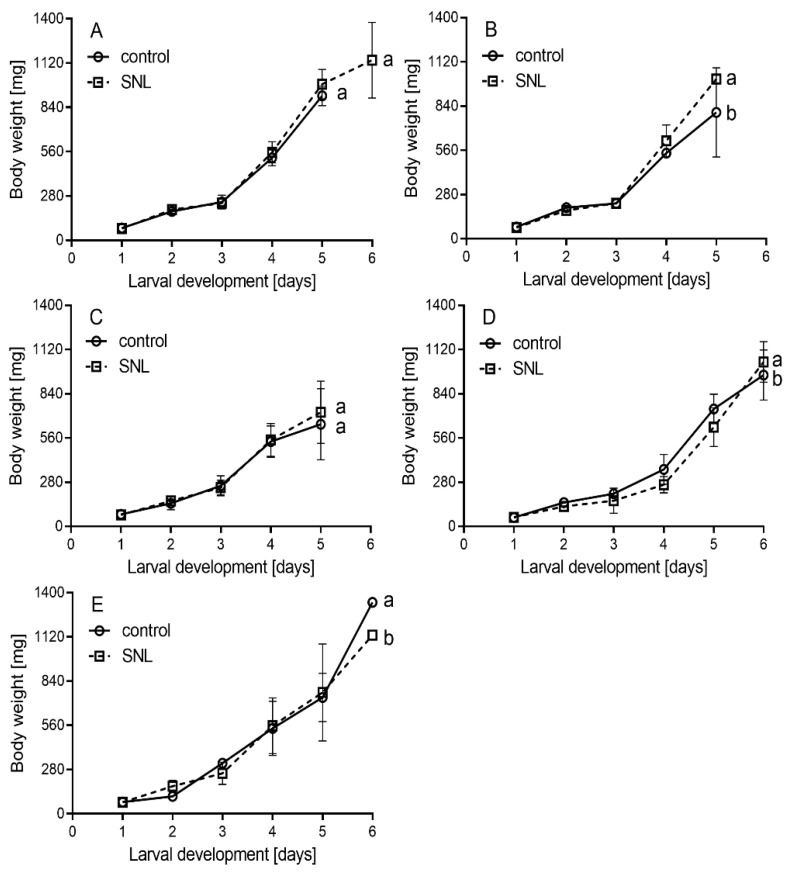
The body weight gain of *S. littoralis* (mean ± SD) larvae in N (**A**), NRC (**B**), CU (**C**), SE (**D**), and SF (**E**) populations in bioassay 3 with GE potato SNL expressing Cry3Aa and control potato Superior. The same letters denote non-significant differences, while different letters denote statistically significant differences. The values of statistical tests are available in Appendix A. N—population of *S. littoralis* reared in our laboratory; NRC—population of *S. littoralis* obtained from National Research Centre, Egypt; CU—population of *S. littoralis* obtained from Cairo University, Egypt; SE—sensitive population of *S. littoralis* obtained from Egypt; SF—sensitive population of *S. littoralis* obtained from France.

**Table 1 plants-11-01312-t001:** Examined parameters (mean ± SD) of three *S. littoralis* populations in bioassay 2 with 8 µg/g recombinant and natural Cry3Aa in a semi-artificial diet. Different letters (in bold) denote statistically significant differences, while letters are not assigned when statistical difference was not found. Statistical comparison was performed for recombinant and natural Cry3Aa together. The values of statistical tests are available in Appendix A.

Population	Examined Parameter	Control	Recombinant Cry3Aa	Natural Cry3Aa
NRC	Initial larval weight (mg)	59.1 ± 3.3	60.3 ± 4.0	59.2 ± 3.5
	ECI (%) ^1^	24.2 ± 1.7	36.5 ± 1.9	31.8 ± 11.3
	Weight increment (mg)	950.0 ± 176.8 **b**	1077.7 ± 149.5 **a**	945.8 ± 153.1 **b**
	Maximal body weight (mg)	1009.1 ± 176.4 **b**	1137.9 ± 148.7 **a**	1004.9 ± 152.9 **b**
	Length of fifth instar (days)	2.9 ± 0.4	2.9 ± 0.3	2.9 ± 0.3
	Length of sixth instar (days)	2.9 ± 0.4	2.8 ± 0.4	2.9 ± 0.4
	Length of prepupal stage (days)	2.6 ± 0.6	2.7 ± 0.5	2.5 ± 0.5
	Larval mortality (%)	0	0	7.5
	Pupal weight (mg)	363.4 ± 52.5 **b**	386.9 ± 45.9 **a**	366.2 ± 40.4 **b**
	Length of pupal stage (days)	8.7 ± 0.7 **b**	10.2 ± 0.7 **a**	8.5 ± 0.5 **c**
	Pupal mortality (%)	0	0	8.1
	No. of laid eggs per female per day	281.2 ± 125.6	294.7 ± 121.3	274.6 ± 127.8
	No. of hatched eggs per female per day	136.1 ± 79.4	140.2 ± 127.9	147.1 ± 108.3
	Hatching rate per female per day (%)	43.8 ± 17.0	36.9 ± 30.4	50.2 ± 22.0
SE	Initial larval weight (mg)	58.6 ± 3.6	59.7 ± 3.2	58.0 ± 2.8
	ECI (%)	25.0 ± 7.5	28.7 ± 1.8	20.2 ± 3.8
	Weight increment (mg)	825.6 ± 104.8 **a**	800.1 ± 114.6 **a**	709.7 ± 177.0 **b**
	Maximal body weight (mg)	884.3 ± 103.4 **a**	859.8 ± 114.0 **a**	767.7 ± 176.4 **b**
	Length of fifth instar (days)	2.6 ± 0.5 **a**	2.2 ±0.4 **b**	2.6 ± 0.5 **a**
	Length of sixth instar (days)	2.9 ± 0.4	3.1 ± 0.5	3.1 ± 0.5
	Length of prepupal stage (days)	2.5 ± 0.5	2.4 ± 0.6	2.7 ± 0.8
	Larval mortality (%)	20.0	32.5	32.5
	Pupal weight (mg)	327.2 ± 42.3 **a**	303.9 ± 43.8 **ab**	299.0 ± 52.9 **b**
	Length of pupal stage (days)	8.5 ± 0.7	8.2 ± 0.9	8.3 ± 0.8
	Pupal mortality (%)	6.3	0	3.7
	No. of laid eggs per female per day	298.9 ± 132.5	328.8 ± 148	252.2 ± 123.9
	No. of hatched eggs per female per day	256.9 ± 150.3	292.2 ± 170.8	214.3 ± 135.5
	Hatching rate per female per day (%)	75.8 ± 29.5	81.8 ± 23	72.9 ± 34.9
SF	Initial larval weight (mg)	59.9 ± 3.1	60.2 ± 3.5	59.3 ± 3.3
	ECI (%)	21.4 ± 2.8	22.2 ± 3.4	23.2 ± 1.8
	Weight increment (mg)	861.8 ± 211.9	822.9 ± 131.3	811.1 ± 138.0
	Maximal body weight (mg)	921.7 ± 212.2	883.3 ± 131.2	870.4 ± 138.7
	Length of fifth instar (days)	2.3 ± 0.5	2.2 ± 0.4	2.2 ± 0.4
	Length of sixth instar (days)	3.1 ± 0.4	3.1 ± 0.5	3.0 ± 0.4
	Length of prepupal stage (days)	2.7 ± 0.5	2.9 ± 0.5	2.8 ± 0.5
	Larval mortality (%)	17.5	15.0	25.0
	Pupal weight (mg)	330.5 ± 46.1	326.6 ± 39.1	317.4 ± 41.7
	Length of pupal stage (days)	8.2 ± 0.8 **ab**	8.1 ± 0.7 **b**	8.6 ± 1.3 **a**
	Pupal mortality (%)	15.2	11.8	3.3
	No. of laid eggs per female per day	245.8 ± 152.1	334.4 ± 180.6	259.0 ± 133.2
	No. of hatched eggs per female per day	157.1 ± 118.9	227.3 ± 179.6	165.9 ± 120.2
	Hatching rate per female per day (%)	48.5 ± 30.1	61.0 ± 29.8	52.4 ± 32.7

^1^ ECI: efficiency of food conversion to biomass.

**Table 2 plants-11-01312-t002:** Examined parameters (mean ± SD) of fiv*e S. littoralis* populations in bioassay 3 with Cry3Aa expressed in GE potato SNL. Different letters (in bold) denote statistically significant differences, while letters were not assigned when statistical difference was not found. The values of statistical tests are available in Appendix A. Abbreviation in the table—for details, see Section 5, Materials and Methods.

Population	Examined Parameter	Control	GE Potato SNL
N	Initial larval weight (mg)	75.7 ± 2.4	74.7 ± 2.5
	ECI (%) ^1^	24.1 ± 12.1	23.1 ± 5.1
	Weight increment (mg)	964.6 ± 141.5	933.3 ± 170.5
	Maximal body weight (mg)	1040.3 ± 141.6	1008.0 ± 170.5
	Length of fifth instar (days)	2.4 ± 0.5	2.2 ± 0.4
	Length of sixth instar (days)	3.5 ± 0.6	3.4 ± 0.6
	Length of prepupal stage (days)	1.7 ± 0.6	1.9 ± 0.6
	Larval mortality (%)	33.3	43.3
	Pupal weight (mg)	341.3 ± 30.2	326.1 ± 43.3
	Length of pupal stage (days)	10.4 ± 0.7	10.4 ± 0.6
	Pupal mortality (%)	0^2^	0
	No. of laid eggs per female per day	217.3 ± 96.6	260.0 ± 142.5
	No. of hatched eggs per female per day	144.4 ± 130.3	150.6 ± 147.6
	Hatching rate per female per day (%)	60.8 ± 21.5	46.9 ± 29.6
NRC	Initial larval weight (mg)	72.7 ± 2.4	71.7 ± 1.8
	ECI (%)	24.0 ± 5.7	33.1 ± 16.1
	Weight increment (mg)	872.7 ± 125.6	913.6 ± 109.4
	Maximal body weight (mg)	945.4 ± 125.2	985.3 ± 110.0
	Length of fifth instar (days)	2.0 ± 0^2^	2.0 ± 0
	Length of sixth instar (days)	3.4 ± 0.6 **a**	3.0 ± 0 **b**
	Length of prepupal stage (days)	2.6 ± 0.5	2.4 ± 0.5
	Larval mortality (%)	36.7	53.3
	Pupal weight (mg)	333.3 ± 17.8	340.1 ± 25.5
	Length of pupal stage (days)	8.2 ± 0.4 **b**	9.4 ± 0.7 **a**
	Pupal mortality (%)	0^2^	0
	No. of laid eggs per female per day	297.5 ± 70.4	376.2 ± 152.7
	No. of hatched eggs per female per day	232.8 ± 94.3	262.7 ± 159.1
	Hatching rate per female per day (%)	74.5 ± 20.2	67.9 ± 31.2
CU	Initial larval weight (mg)	74.5 ± 3.3	73.9 ± 3.3
	ECI (%)	17.9 ± 4.2	19.8 ± 4.7
	Weight increment (mg)	703.5 ± 204.7	740.6 ± 138.9
	Maximal body weight (mg)	778.0 ± 203.5	814.5 ± 138.0
	Length of fifth instar (days)	2.0 ± 0.2	2.1 ± 0.3
	Length of sixth instar (days)	3.0 ± 0.2	2.9 ± 0.4
	Length of prepupal stage (days)	1.7 ± 0.5	1.9 ± 0.3
	Larval mortality (%)	40.0	46.7
	Pupal weight (mg)	287.4 ± 39.9	288.3 ± 42.6
	Length of pupal stage (days)	8.1 ± 0.9	7.6 ± 0.8
	Pupal mortality (%)	0^2^	0
	No. of laid eggs per female per day	348.6 ± 108.3	300.4 ± 73.1
	No. of hatched eggs per female per day	321.9 ± 110.1	217.1 ± 78.0
	Hatching rate per female per day (%)	91.8 ± 4.0	71.5 ± 14.9
SE	Initial larval weight (mg)	60.8 ± 6.8	61.6 ± 6.6
	ECI (%)	24.2 ± 4.0	25.6 ± 4.1
	Weight increment (mg)	868.3 ± 145.	912.2 ± 148.4
	Maximal body weight (mg)	929.1 ± 142.8	973.8 ± 149.9
	Length of fifth instar (days)	2.3 ± 0.7	2.3 ± 0.5
	Length of sixth instar (days)	3.8 ± 0.8	3.8 ± 0.5
	Length of prepupal stage (days)	1.4 ± 0.5	1.6 ± 0.5
	Larval mortality (%)	6.7	16.7
	Pupal weight (mg)	291.0 ± 52.2	298.8 ± 47.9
	Length of pupal stage (days)	8.8 ± 0.8 **b**	9.3 ± 0.9 **a**
	Pupal mortality (%)	3.6	8.3
	No. of laid eggs per female per day	322.9 ± 134.2	317.8 ± 76.9
	No. of hatched eggs per female per day	194.1 ± 94.6	161.7 ± 60.2
	Hatching rate per female per day (%)	60.2 ± 16.4	51.0 ± 17.7
SF	Initial larval weight (mg)	73.6 ± 3.8	74.3 ± 3.9
	ECI (%)	16.7 ± 3.6	21.0 ± 7.7
	Weight increment (mg)	769.9 ± 324.3	973.8 ± 294.2
	Maximal body weight (mg)	843.5 ± 325.1	1048.1 ± 293.0
	Length of fifth instar (days)	1.4 ± 0.5 **a**	1.1 ± 0.3 **b**
	Length of sixth instar (days)	3.8 ± 0.9	4.4 ± 1.0
	Length of prepupal stage (days)	1.3 ± 0.4 **b**	1.4 ± 0.5 **a**
	Larval mortality (%)	6.7	3.3
	Pupal weight (mg)	295.0 ± 81.1	351.1 ± 92.2
	Length of pupal stage (days)	10.7 ± 0.8 **a**	10.1 ± 0.8 **b**
	Pupal mortality (%)	57.1 **a**	24.1 **b**
	No. of laid eggs per female per day	332.9 ± 203.7	255.7 ± 146.7
	No. of hatched eggs per female per day	114.1 ± 145.4	150.9 ± 149.0
	Hatching rate per female per day (%)	32.2 ± 36.4	41.4 ± 39.8

^1^ ECI: efficiency of food conversion to biomass. ^2^ Statistical comparison is impossible because of no variability in data. N—population of *S. littoralis* reared in our laboratory; NRC—population of *S. littoralis* obtained from National Research Centre, Egypt; CU—population of *S. littoralis* obtained from Cairo University, Egypt; SE—sensitive population of *S. littoralis* obtained from Egypt; SF—sensitive population of *S. littoralis* obtained from France.

## Data Availability

All published data are available within the article and Appendix A.

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
