# Peer review of "Cry3Aa Toxin Is Not Suitable to Control Lepidopteran Pest Spodoptera littoralis (Boisd.)"

_plants, 2022, doi:10.3390/plants11101312_

Round 1

Reviewer 1 Report

The manuscript presented by the Authors reports an interesting assessment of the possible susceptibility of a lepidopteran phytophagous pest (Spodoptera littoralis) to a coleopteran-specific Bacillus thuringiensis Cry toxin (Cry3Aa). As the Authors stated in their introduction, there have been reports of some sub-lethal effects of the Cry3Aa toxin when tested on S. littoralis, so I find interesting and useful their effort to extend the existing data regarding the efficacy of Cry3Aa toxin on S. littoralis.

I found that only some minor text editing is needed before publication.

Line 61: change “agens” in “agents”;

Line 100: change “ug” in “μg”;

From line 178 (3. Discussion) to the end of manuscript: change the character of latin names using italic (“S. littoralis” in “S. littoralis”, “L. decemlineata” in “L. decemlineata”, “Manduca sexta” in Manduca sexta”, “Bacillus thuringiensis ssp. kustaki” in Bacillus thuringiensis ssp. kustaki”, “Ricinus communis” in Ricinus communis”, “Escherichia coli” in Escherichia coli”, B. thuringiensis ssp. tenebrionisin B. thuringiensis ssp. tenebrionis);

Line 198: change “synthesised” in “synthetised”;

Line 421-422: “body weight gain” is repeated;

Line 476-477: change “includeing” in “including”

Author Response

Response to Reviewer 1 Comments

Dear reviewer,

Thank you very much for your valuable review, advice, and comments. We have explained and accepted all of them and improved the MS accordingly.

Comments and Suggestions for Authors

The manuscript presented by the Authors reports an interesting assessment of the possible susceptibility of a lepidopteran phytophagous pest (Spodoptera littoralis) to a coleopteran-specific Bacillus thuringiensis Cry toxin (Cry3Aa). As the Authors stated in their introduction, there have been reports of some sub-lethal effects of the Cry3Aa toxin when tested on S. littoralis, so I find interesting and useful their effort to extend the existing data regarding the efficacy of Cry3Aa toxin on S. littoralis.

I found that only some minor text editing is needed before publication.

Line 61: change “agens” in “agents”;

Response 1: The mistake was corrected.

Line 100: change “ug” in “μg”;

Response 2: The mistake was corrected.

From line 178 (3. Discussion) to the end of manuscript: change the character of latin names using italic (“S. littoralis” in “S. littoralis”, “L. decemlineata” in “L. decemlineata”, “Manduca sexta” in “Manduca sexta”, “Bacillus thuringiensis ssp. kustaki” in “Bacillus thuringiensis ssp. kustaki”, “Ricinus communis” in “Ricinus communis”, “Escherichia coli” in “Escherichia coli”, “B. thuringiensis ssp. tenebrionis” in “B. thuringiensis ssp. tenebrionis”);

Response 3: It was corrected. It was probably caused by any kind of manipulation with manuscript during submission.

Line 198: change “synthesised” in “synthetised”;

Response 4: The mistake was corrected

Line 421-422: “body weight gain” is repeated;

Response 5: The word was deleted

Line 476-477: change “includeing” in “including”

Response 6: The mistake was corrected

Reviewer 2 Report

I found this manuscript interesting in that it:

1) Presents essentially only negative results

2) Those negative results appear to contradict three previous reports from the same laboratory/authors.

While it is important to present negative results there are many ways in which this manuscript can be improved.

1) The title strongly implies that Cry3Aa may have potential for the control of the pest. The title should be honest and state that Cry3Aa has no potential.

2) The authors do not have a strong understanding of Bt Cry proteins. For example in line 33 they state that Cry toxins interact with cadherins. While a few might most interact with other receptors such as APNs and ABC proteins.

3) Several other statements in the introduction are too broad/vague. Eg when talking about "unique modes of activation". This reviewer has no idea what this means.

4) Having said that Cry1B is active against three orders of insect the manuscript then states that Cry1 toxins are specifically active against Lepidoptera!

5) I don't understand the purpose of Fig 1. If they are just demonstrating that the two Cry3Aa preps have different concentrations then we really don't need to see those data. However having shown Fig 1 they need to explain why there are two peaks for Cry3Aa. Was the concentration based on integrating both peaks?

6) The manuscript states that there was a significant difference between the native and recombinant Cry3Aa preps. We are given LD90 values but no indication of error limits. A probit analysis was used and so traditionally the results are expressed as LC50 values with corresponding FLs. These need to be provided.

7) Calculated LC50 values should be compared with those in the literature for Cry3Aa and Sl.

8) I really don't see the need to present the graphs for Fig 2. Just do the number crunching and provide LC50 values.

9) I find the long list of tabulated data in Tables 1 and 2 very off putting. These should be presented in graphical form (bar graphs)

10) It is not clear if the Cry3Aa protein was produced as crystals or silubilized protein in E.coli. For both sources it was 'diluted' in potassium bicarbonate. Was this designed to solubilize the crystals?

11) The authors attempt to explain the difference between native and recombinant Cry3Aa but there are many unanswered questions such as do the two proteins have the same sequence? Are they presented as crystals or solubilized protein? To confirm the calculated protein concentration I would really like to see an SDS PAGE gel.

12) There are many instances in the manuscript where species names have not been italicized.

Author Response

Response to Reviewer 2 Comments

Dear reviewer,

Thank you very much for your valuable review, advice, and comments. We have explained and/or accepted all of them and improved the MS accordingly.

Comments and Suggestions for Authors

I found this manuscript interesting in that it:

1) Presents essentially only negative results

2) Those negative results appear to contradict three previous reports from the same laboratory/authors.

While it is important to present negative results there are many ways in which this manuscript can be improved.

Thank you for your above-mentioned comments.

1) The title strongly implies that Cry3Aa may have potential for the control of the pest. The title should be honest and state that Cry3Aa has no potential.

Response 1: The title of the manuscript was changed.

2) The authors do not have a strong understanding of Bt Cry proteins. For example in line 33 they state that Cry toxins interact with cadherins. While a few might most interact with other receptors such as APNs and ABC proteins.

Response 2: We have simplified and modified the text.

3) Several other statements in the introduction are too broad/vague. Eg when talking about "unique modes of activation". This reviewer has no idea what this means.

Response 3: The text was rewritten and clarified. The problematic sentence was deleted because it was not important for our study.

4) Having said that Cry1B is active against three orders of insect the manuscript then states that Cry1 toxins are specifically active against Lepidoptera!

Response 4: The sentence was modified. We apologize for the inaccuracies.

5) I don't understand the purpose of Fig 1. If they are just demonstrating that the two Cry3Aa preps have different concentrations then we really don't need to see those data. However having shown Fig 1 they need to explain why there are two peaks for Cry3Aa. Was the concentration based on integrating both peaks?

Response 5: HPLC analysis demonstrated that the same compounds are present in both recombinant and natural Cry-3Aa solutions, according to the similar elution profiles, particularly in the key area – see the elution time of 15-20 minutes. We do not know why two peaks appear in both profiles at the moment, however, we are convinced this is not important for our study: a confirmation we were working with the same compounds is sufficient for our experiments. Unfortunately, analytical analysis of the Cry-3Aa protein(s) is beyond the scope of this paper.

See also point 11.

6) The manuscript states that there was a significant difference between the native and recombinant Cry3Aa preps. We are given LD90 values but no indication of error limits. A probit analysis was used and so traditionally the results are expressed as LC50 values with corresponding FLs. These need to be provided.

Response 6: We presented LC90 because subsequently in bioassay 2 we work with concentration corresponding to LC90 of recombinant Cry3Aa. Confidence intervals as well as LC50 were added.

7) Calculated LC50 values should be compared with those in the literature for Cry3Aa and S. l.

Response 7: The comparison was added to Discussion.

8) I really don't see the need to present the graphs for Fig 2. Just do the number crunching and provide LC50 values.

Response 8: Yes, simply presenting the LC50 values would suffice; but, we prefer the graph presentation because it is far more illustrative.

9) I find the long list of tabulated data in Tables 1 and 2 very off putting. These should be presented in graphical form (bar graphs)

Response 9: Tables 1 and 2 are, without a doubt, quite long. However, the appropriate presentation of their data is a question. We suppose that utilizing graphs would be much more baffling than using tables. Therefore, we prefer to use tables.

10) It is not clear if the Cry3Aa protein was produced as crystals or solubilized protein in E. coli. For both sources it was 'diluted' in potassium bicarbonate. Was this designed to solubilize the crystals?

Response 10: Both toxins were produced as crystals. This information was added into M&M section.

11) The authors attempt to explain the difference between native and recombinant Cry3Aa but there are many unanswered questions such as do the two proteins have the same sequence? Are they presented as crystals or solubilized protein? To confirm the calculated protein concentration I would really like to see an SDS PAGE gel.

Response 11: Yes, recombinant and natural proteins share the same primary amino acid sequence, and HPLC validated the identical components in both solutions of proteins, as shown in Fig. 1. The amounts of Cry3Aa in these solutions were also measured using a particular ELISA, which is a very sensitive approach ideal for this purpose. Sorry, but SDS PAGE is only a semi-quantitative procedure and is completely inappropriate when far more precise analytical methods (like the ones we used) are available.

12) There are many instances in the manuscript where species names have not been italicized.

Response 12: It was corrected. It was probably caused by any kind of manipulation with manuscript during submission.

Round 2

Reviewer 2 Report

The authors have added much of the additional information and data that I requested but I feel that there is still some confusion over the methodology used which may or may not impact on the one result of the manuscript that particularly interests me - the difference in activity (to CPB) between native and recombinant Cry3Aa.

It has been clarified that both forms of Cry3Aa were provided as crystals. Section 5.4 states that both crystal preps were solubilized in potassium hydrogen carbonate but does not state whether non-solubuilized material was included in the bioassays. In other words was the solution centrifuged and the supernatent removed at any point? For the HPLC experiment it is stated that the toxin was dissolved in TFA, from which we can infer that crystals were used rather than the pre-dissolved toxin in the working solution. Do Cry3Aa crystals dissolve in TFA? The authors dismiss SDS-PAGE as a quantification method - which is fine, but it can still serve as a good sanity check as to what exactly is being measured. It may well be a co-incidence that despite the initial crystal concentration reportedly being similar the working concentrations show a 7.5 fold difference which is similar to the 6.8 fold difference in activity. 

Other minor points. 

1) The comment that Cry toxins interact mainly with cadherins simply can't be justified. Some Cry toxins do interact with cadherins but most almost certainly do not.

2) When stating that the native Cry3Aa is 6.8 times more toxic than the recombinant it should be clarified that this is referring to the fact that the LC90 values differ by 6.8 fold. The LC50 values vary by 18 fold - you can't give a single figure for the difference, although most researchers use LC50.
